# Working Carers in Europe and How Their Caring Responsibilities Impact Work–Family Life Conflict: Analysis of the European Quality of Life Survey

**DOI:** 10.3390/healthcare12232415

**Published:** 2024-12-02

**Authors:** Valentina Hlebec, Miriam Hurtado Monarres, Zdenka Šadl

**Affiliations:** Faculty of Social Sciences, University of Ljubljana, Kardeljeva pl. 5, 1000 Ljubljana, Slovenia; miriam.hurtado-monarres@fdv.uni-lj.si (M.H.M.); zdenka.sadl@fdv.uni-lj.si (Z.Š.)

**Keywords:** work–life conflict, working carers, informal care, family care, formal care services, care regimes

## Abstract

**Background/Objectives**: Ageing of the population is a pertinent characteristic in developed societies that raises questions of who provides care and how care is provided to frail and dependent old people. The majority of care is provided by family members, friends, and neighbours, many of whom are of working age and active in the labour market. The aim of this study is to research how work and care are intertwined and how they cause conflict for individuals in Europe. **Methods:** A hierarchical regression analysis of European Quality of Life Survey data was performed to evaluate the amount of explained variance of work–life conflict according to caring situation, working conditions, and demographic characteristics of an adult European population. A stratified, clustered multistage sample design was used to select 15,656 adult respondents working as employees. **Results:** The results show that the three blocks explain about 18% of work–life conflict, with working conditions being the most influential block, followed by demographic characteristics and caring situation. **Conclusions:** The frequency of caring, use of formal care, and quality of formal services significantly mitigate work–life conflict, together with the number of working hours, commuting, fear of losing one’s job, fear of having insufficient income in old age, and the ease of making ends meet. Care regimes also have a considerable effect on work–life conflict. Countries have the power and responsibility to support working carers in their multiple and often conflicting roles by allowing flexible work arrangements, the right to reduce the number of working hours safely, employment protection during care, emergency leave, and short- and long-term leave, as well as by investing in community-based long-term care models and services.

## 1. Introduction

With the increase in life expectancy and an ageing population, the demand for caregiving for dependant older people is rising. Most often, care is provided by people closest to the older person, namely relatives, neighbours, and friends. At the same time, the time that informal carers need to stay active in their employment is increasing, which causes conflicts between their work and life responsibilities. In the following text, we investigate the informal carers of older people, the difference between working and non-working carers, and the gender gap in care responsibilities. We then show how work and care are intertwined and the effects they have on working carers, dimensions of work–life conflicts, why caregivers remain engaged with the workplace despite the challenges in balancing care and paying work, and how organisations and governments help in supporting caregivers in their roles.

In Europe [1], 12% of the adult population provides care for disabled or infirm family members, neighbours, or friends under 75 years of age, and 12% provides care to disabled or infirm family members, neighbours or friends who are older than 75 years of age. Among the working population aged 18–64, 88% are non-carers, 7% are working carers, and 5% are carers not in employment. A gender gap persists, notably in everyday caring (twice as many women as men). A comparison of working and non-working carers seems to show that working carers are better-off than non-working carers, as they are in better health, better-off financially, and less lonely, in addition to scoring lower values for the social exclusion index and reporting higher life satisfaction. Although research confirms the listed advantages and benefits of paid employment for informal carers [2], the difficulties they face while combining care and work can lead to increased sick leave or mental health issues, such as greater stress and depression [3].

Indeed, there are two competing perspectives when it comes to conceptual frameworks for understanding how work and care are intertwined and the effects they have on working carers. The first one stresses enrichment and accumulation, as well as a positive flow between the worlds of work and caring, whereas the second one emphasises overload, strain, and burden. Beigi and Shirmohammadi [4] cite Kahn et al.’s 1964 role theory that says that the positive perspective employs terms like role enhancement [5,6], role accumulation [7], and enrichment, all of which underscore the positive enhancing effects of intertwining multiple roles. The negative perspective points to role strain/overload, also known as the depletion or scarcity hypothesis, and underlines the negative relationship between working and caring [8], leading to the recognition that, in many respects, the requirements of work and caring are incompatible. Given three dimensions of work–life conflicts, i.e., a lack of time (the time requirements of work or care prevent fulfilment of the second role), exhaustion (one role takes so much effort that it prevents the second one from being fulfilled), and norms and behaviours (each sphere assumes a different culture, norms, and behaviours that are non-transferable, namely being a good mother or daughter is not the same as being a good worker), we understand that many carers find it difficult to combine work and care. The conflict between work and care, especially for women, is manifested through employment transitions (from full- to part-time work or withdrawal from the labour market and from part-time work to withdrawal from the labour market) [9], which depend on the intensity of family care, the location of care (the caregiver and care recipient living together or in separate households), the family caregiving relationship, and the various care tasks involved [9]. However, the relationship can also work in the opposite direction; changes in work demands may lead to a lower likelihood of providing (intensive) care.

Some studies [10,11] have shown that increasing the demands of caring in terms of hours of work and perceived care burden for a dependent old family member adds to work strain, which is also exacerbated by job demands; a lack of support from supervisors and co-workers; and little control over how, when, and where work is performed. On the other hand, Stewart [12] observed parents with children with a disability and parents of children without a disability and found similar factors increasing work–family life conflict, although they also proposed policies for organisations like flexible work measures, which relieve such conflict. Li et al. [13] showed that support from family and in the workplace is important for easing work–life conflict for carers of people with disabilities.

The decision to remain engaged with the workplace from the onset of care need is, as suggested in [11], linked to financial security and subject to perceptions that the workplace culture will support employees with caring responsibilities. However, Ehrlich [9] established that women’s employment behaviour during family caregiving depends on the household income, marital status, and women’s relative economic position within the household, alongside the varying intensity of family caregiving (i.e., the time committed to caregiving). Apart from financial constraints, there are also other work-related factors, such as person–job fit, as well as the physiological and psychological requirements of the job itself and gender differences [14], that must be considered while designing policy support for working carers. Even though working carers might envisage maintaining their workplace activity, actually remaining in the workplace depends on a combination of factors like a family care-friendly working environment, supportive supervisors and co-workers, and the existence of an array of flexibility measures. Constantin et al. [15] showed that relationships exist between care hours and care strain and between care hours and care strain and withdrawal from contractually provided working hours and withdrawal from career development and progression. Care strain was found to have a stronger relationship with all types of work withdrawal than hours of care but was moderated by a family care-supportive work environment, demonstrating the importance of both the work culture and organisational measures that can allow working carers to remain in the labour market, together with a supportive family environment. Similarly, previous study [12] explored both work culture and the availability of flexible (‘flexi’) work measures, as well as control over where and when work is carried out and support from co-workers and supervisors. It seems that the decision to remain active in the labour market is not only shaped by an individual with caring responsibilities but also by the context in which work is performed and the ways in which work can be carried out. Murphy and Cross [16] indicated that establishing and keeping a balance and borders between work and care responsibilities rely heavily on supervisors’ understanding of the caring situation, encouraging working carers to ask for and take advantage of organisational or institutional measures. Among policy measures, remote working and flexi hours or compressed hours were found to be most often utilised against carers’ leave or force majeure leave in Ireland, as carers were either unaware of what they were and how eligibility was established and were even reluctant to explain their personal family situation to HR managers. Similarly, an earlier study [17] was able to cover 50 organisations and found that good work outcomes were linked to supportive supervisors and co-workers and family care-friendly organisations. A review [18] showed similar results in terms of the need for support from both supervisors and organisations, as well as from national policies, regardless of whether the caring responsibilities were for a child or an older person, particularly underlining the need to support women’s labour participation in countries with traditional gender roles. The greater involvement of women in care compared to men [19] may lead to more critical situations for employed women when they start to provide (more intensive) care, i.e., a greater risk of time pressure and, in turn, a higher likelihood of experiencing time conflict and strain, consequently reducing their working hours or causing them to leave the labour market altogether [9]. Even in countries with more progressive gender roles, women are often pressured to compromise their work roles, which can be detrimental to their labour market careers and economic independence in the long run. Therefore, as [20] points out using the example of the Netherlands, while designing care-friendly workplaces, special attention should be paid to women’s needs and preferences, since they are more prone to reducing their working hours than men. According to some findings [3], the protective effects of workplace support are greater for women than for men. It seems that women (working and caring) need more support from the organisational environment [13]. As a result of conflict between work and care and an unsupportive workplace, working carers may also change jobs or become self-employed as two additional exit options, which preserve attachment to the labour market but in a more sustainable way than working part-time or leaving the labour market altogether [21,22].

Mullins et al. [23] observed public sector organisations and determined that flexibility measures such as condensed work hours, a flexible beginning and end of work hours, and working from home significantly reduced conflict between work and family life. The positive effect of flexibility and support in the workplace was also confirmed by the authors of [3], who found that significant flexibility at work, when compared with no flexibility, and high levels of supervisor and co-worker understanding of employees’ caring responsibilities (compared with no understanding in both cases) were associated with lower scores for work–care conflict. Czerwińska-Lubszczyk and Byrtek [24] established that larger companies, in contrast to small or medium-sized companies, have a wider array of support measures applied and that work–life balance was reported as higher in larger companies than in smaller ones. Kossek et al. [25] observed that policy measures, ranging from organisational to national, delineate types of control over work–life boundaries (spatial, size, temporal, permeability, continuity) and that policy measures must be available, attainable and work in a specific organizational and institutional culture. Positive outcomes for individuals are, hence, only obtainable when all of these factors act synergistically.

New laws or regulations also play an important role in mitigating the conflict between work and care. Monks [26] states that family-friendly policies on care and work, including working time policies and policies aimed at developing leave regulations, contribute to improving work–life balance. In Europe, new initiatives have been adopted in recent years [27,28,29] to support work–life balance for carers, albeit their implementation has many facets depending on the historical development of specific country care policies (e.g., [30,31,32]). Similarly, in other developed countries, policies are currently being observed and studied [33].

How do organisations and institutions support working carers and what do working carers need in order to combine work and care simultaneously? A policy study conducted by EUROFOND [34] structured a carer’s needs in line with Maslow’s pyramidal hierarchy of needs. The basic needs of working carers are flexible work-time arrangements (and long-term work accounts), followed by the right to reduce the number of working hours safely by taking reversible (safe), part-time work and employment protection during care, as well as emergency leave and short- and long-term leave (unpaid and/or paid). Next, they collected policy measures offered to working carers across European countries and devised three groupings of countries according to the richness of policy offerings to working carers: fully fledged care regimes based on extended leave entitlements, work flexibility, and protection (DK, DE, FI, BE, SE, FR, AT, IE, LT, and UK); partial care regimes with short-term leave entitlements and protection of working carers (IT, NO, SI, LU, HHR, and EE); and residual care regimes mainly based on flexible work (time) organisation only (CY, RO, PT, C, NL, ES, MT, BG, SKK, EL, LV, U, and PL).

The purpose of this study is to show how work and care are intertwined and how they cause conflict for individuals in Europe. We performed this by taking the demographic characteristics of adult residents of Europe and the country (policy) context into account, together with the care and work context of respondents revealed by EQLS (European Quality of Life Survey) data for 2016. Here, we demonstrate the workplace characteristics at the micro, meso, and macro levels essential to support family carers in continuing to participate in the labour market simultaneously or consecutively while providing family care.

## 2. Materials and Methods

This study utilises data from the European Quality of Life Survey 2016 [35]. The survey has been conducted every 4 years since 2003, except for 2020, owing to the COVID-19 pandemic. The target population is the adult population aged 18 years and over, whose usual place of residence is in the territory of the surveyed country. Since the studied population is found in different countries, the sample size is set at a minimum 1000 per country, with some exceptions with a larger sample size. A stratified, clustered multistage sample design was used to select respondents. Random probability sampling was applied at each stage of sample selection ([35], p. 119). The final stage was the selection of households and eligible individuals within them. The 2016 questionnaire consisted of 104 questions and 262 items, with special attention paid to public services in this wave. The TRAPD procedure was used in translation [36], with cognitive testing applied in the pretesting of the questionnaire. Data were collected via a face-to-face data collection mode with CAPI technology. The WCalib_crossnational_total weight was used to weigh the data, as country subgroupings were employed as an independent variable in regression analysis. Only those working as an employee were selected, since adult working people could have developed a conflict in relation to their supervisors and coworkers. Further, only working people with at least 20 h of work per week were included in the analysis.

The dependent variable is the work–life conflict index, calculated as the sum of the following three items measured in terms of frequency (1, never; 2, less often or rarely; 3, several times a year; 4, several times a month; 5, several times a week; 6, every day):-I have come home from work too tired to do some of the household jobs which needed to be done;-It has been difficult for me to fulfil my family responsibilities because of the amount of time I spend at work;-I have found it difficult to concentrate at work because of my family responsibilities.

An increasing index value means greater work–life conflict. Cronbach’s Alpha was 0.784. Note: Only one item measures the spillover from family to work.

Independent variables are presented in three segments: control and country context (Table 1), care-related items (Table 2), and work-related items (Table 3).

Given that care characteristics and work situation conceptually contribute to work–life conflict as two separate sources, apart from the control variables described in the Introduction, the hierarchical regression method was used. The control block was entered first, the care-related block was entered second, and the work-related block was entered third. All analyses were performed using IBM SPSS 29.0.0.0.

## 3. Results

The average level of work–life conflict was 3.02. The analysed sample consisted of 45.4% female respondents with an average age of 41 years, 58.3% of whom were married, living in households of 3 people, on average. In general, health was evaluated at 1.94. A proportion of 7.6% of respondents lived in open countryside, 38.1% in a village or small town, 23.1% in a medium to large town, and 31.2% in a city or city suburb. A proportion of 13.0% lived in a PCR with short-term leave entitlements and protection for working carers, whereas 34.0% lived in an RCR mainly based on flexible work (time) organisation only.

Frequency of care was evaluated at 3.00, on average; 9.6% received care from family members, friends, or neighbours free of charge; 3.9% received paid care from someone outside of formal health and care services; 6.0% used nursing care services at the home; and 5.2% used home help or personal care services in the home. The quality of long-term care services was estimated at 6.15, on average.

A proportion of 29% of respondents were working in the public sector, with an average likelihood of losing one’s job in the next 6 months of 1.98. The majority (81%) were working on permanent contracts, with 40.44 working hours, on average. The average number of minutes spent commuting was 40.32. On average, fear that income in old age will not be sufficient was evaluated at 1.91, and the ease of making ends meet was estimated at 3.00, on average.

The results of the hierarchical regression analysis are presented in Table 4.

The block of demographic and country context variables that was entered first explains 6% of variability of the work–life conflict index (*p* < 0.001). The R^2^ and adjusted R^2^ have the same value. A number of independent variables have a significant effect on the mentioned index. Namely, men have significantly lower values (b = −0.069; *p* = 0.003; B = −0.030) than women. As age increases, work–life conflict diminishes (b = −0.011; *p* < 0.001; B = −0.112). This is the second-most powerful predictor of work–life conflict. Household size, i.e., having more people in the household, increases work–life conflict significantly (b = 0.073; *p* < 0.001; B = 0.082), making it the third-most influential independent variable. Diminishing health significantly increases work–life conflict (b = 0.331; *p* < 0.001; B = 0.216), making it the most influential independent variable in the first block of predictor variables. The country context was entered as two separate variables compared to a fully fledged care regime, namely a partial care regime and a residual care regime. Both country variables have a significant influence on work–life conflict; a partial care regime increases work–life conflict (b = 0.100; *p* = 0.005; B = 0.030) in a similar way to residual care regime (b = 0.174; *p* < 0.001; B = 0.027). A residual care regime has the fourth-strongest impact on work–life conflict.

The block of variables that was entered second include care variables, which explain a small amount of variance (only 1%; *p* < 0.001). The R^2^ (0.076) and adjusted R^2^ (0.075) remain very similar. The variables entered first, namely the demographic and country context variables, have similar effects on work–life conflict; therefore, we do not interpret them again and only focus on care variables at this stage. The frequency of informal care provided by respondents significantly increases work–life conflict (b = 0.016; *p* = 0.018; B = 0.025), whereas care provided by other family members, friends, or neighbours does not. Paid care received from someone from outside of formal health and care services significantly increases work–life conflict (b = 0.159; *p* = 0.011; B = 0.029), as does nursing care in the home (b = 0.184; *p* = 0.002; B = 0.043). The use of home help or personal care services in the home significantly decreases work–life conflict for respondents (b = −0.283; *p* < 0.001; B = −0.060), as does the quality of long-term services (b = −0.050; *p* < 0.001; B = −0.092). The quality of long-term services has the strongest effect on work–life conflict, followed by the use of home help or personal care services and the use of home nursing services.

The block of variables that was entered third comprises labour market variables, which explain a considerable proportion of variability in the work–life conflict index (10%; *p* < 0.001; R^2^ of 0.177 and adjusted R^2^ of 0.175). Given the explanatory power of this block, the relationships of the variables included in the first two blocks with the dependent variable also changed considerably. The effect of gender is strong (b = −0.172, *p* < 0.001; B = −0.076), the effect of age is a little weaker (b = −0.009, *p* < 0.001; B = −0.085), that of marital status becomes marginally significant (b = 0.048, *p* = 0.056; B = 0.021), and that of household size becomes weaker (b = 0.057, *p* < 0.001; B = 0.064), while the effect of the subjective evaluation of health becomes much weaker (b = 0.207, *p* < 0.001; B = 0.136). The most striking difference occurred for the country context in terms of the regression coefficients changing their size and direction. In particular, a partial care regime has a stronger effect on work–life conflict (b = −0.137, *p* < 0.001; B = −0.041) than a residual care regime (b = −0.061, *p* = 0.020; B = −0.025). In the second block of variables, all variables have a similar, yet smaller effect on work–life conflict, with the exception of frequency of care, which has a stronger effect on work–life conflict (b = 0.021, *p* = 0.001; B = 0.033). The absolute importance of predictor variables in the second block changed with frequency, becoming the third-most important predictor apart from the quality of long-term services and the use of home help or personal services. All of the variables included in the third group of variables, namely work-related variables, have a significant effect on work–life conflict. Being employed in the public sector significantly increases work–life conflict (b = 0.047, *p* = 0.052; B = 0.019), fear of losing one’s job in the next 6 months considerably increases work–life conflict (b = 0.113, *p* < 0.001; B = 0.106), the number of hours spent working at the main job significantly increases work–life conflict (b = 0.026, *p* < 0.001; B = 0.194), the number of minutes spent commuting from home to work significantly increases work–life conflict (b = 0.002, *p* < 0.001; B = 0.061), and worry that one’s income is not going to be sufficient in old age increases work–life conflict significantly (b = 0.020, *p* < 0.001; B = 0.046), while the difficulty of making ends meet with one’s income has a similar effect(b = 0.195, *p* < 0.001; B = 0.206). In terms of strength, the last variable has the strongest effect on work–life conflict, followed by the number of hours worked at the main job and fear of losing one’s job in the next 6 months. When considering all variables in the final model, only the subjective evaluation of health is as important as the work variables, being the third-most important predictor of work–life conflict.

To better understand the changes in the country context when work-related variables entered the regression model, the averages of all independent variables across country regimes were calculated (see Table 5).

Work–life conflict is, on average, the highest in residual care regimes (3.06) and the lowest in fully fledged care regimes (2.82). Those working at least 20 h per week are the youngest, on average, in residual care regimes (40.51) and the oldest in partial care regimes (43.21). Household size, on average, is largest in residual regimes (3.2) and the smallest in fully fledged care regimes (2.81). The subjective evaluation of health and subjective degree of urbanisation are quite similar across care regimes. The frequency of care is highest, on average, in fully fledged care regimes (3.04) and the lowest in residual care regimes (2.64). The involvement of other family members, friends, and neighbours is the highest in fully fledged care regimes (0.13) and the lowest in residual care regimes (0.07). The rate of paying for care in the grey market is the highest in fully fledged care regimes (0.05) and the lowest in residual care regimes (0.02). The use of nursing care is the highest, on average, in fully fledged care regimes (0.09) and the lowest in residual care regimes (0.04). The same is true for home help services. The quality of long-term care is evaluated as the highest in fully fledged care regimes (6.45) and the lowest in residual care regimes (5.8). The rate of working in the public sector is about the same in all care regimes. The fear of losing one’s job is the highest in residual care regimes (2.09) and the lowest in fully fledged care regimes (1.73). The average number of hours spent working at the main job is the highest in residual care regimes (41.76) and the lowest in fully fledged care regimes (38.04). The average number of minutes spent commuting from home to work is the highest in fully fledged care regimes (38.05) and the lowest in residual care regimes (37.52). The fear of not having adequate income in old age is the lowest, on average, in fully fledged care regimes (5.63) and the highest in residual care regimes (6.59). Making ends meet is the easiest, on average, in fully fledged care regimes (2.73) and the most difficult in residual care regimes (3.24).

## 4. Discussion

The results confirm previous studies in the sense that all three blocks of variables are important in explaining variability in work–family life conflict, as found in [10,12,37,38,39,40]. While caring, as such, is not necessarily connected to work–family life conflict, the care burden with respect to care demands in terms of hours and the subjective burden of care [10], coupled with work demands such as working hours, the obligation to replace missing hours, the fear of losing one’s job, and years spent at work, increases work strain and, in our case, work–family life conflict. Gender and health were also found to be indicative of work-related strain in previous studies. Mortensen et al. [40] found that the predictors of long sick leave among women (but not among men) are a combination of a high level of job strain and informal caregiving; therefore, as [41], they found that financial strain, in particular, impacts women caregivers. Moreover, Mortensen et al. [42] determined that low social support for informal caregivers at work presents a higher risk factor for type 2 diabetes. Female informal carers also experience greater work interference while combining paid work and care, as they provide more hours of care (often alone) across more care domains [43]. We confirmed relationships between the three pillars of predictive variables—control variables, care demands, and work demands—variables that were previously found in separate country studies. We can claim that our study supports the negative spillover hypothesis, indicating that both care demands and work demands contribute to work–family conflict.

The context in which care is carried out by working carers was measured by care regimes in our study, allowing us to investigate the cultural environment in which working carers navigate care and work. The data analysis supports the expectation that in countries where working carers are fully supported by nationally adopted policies, their workplace participation can bear higher levels of work–family life conflict. We assume that working carers must either reduce their working hours or leave the formal working labour market sooner in residual and partial care regimes than in fully fledged care regimes. Given the aggregate level of our analysis and that previous micro-level studies focused on a qualitative exploration of working carers or national-level quantitative studies [11,12,14,15,16,18], we tentatively confirm that a supportive national context encourages working carers’ labour market participation. Working in the public sector, as opposed to the private sector, appears to contribute to work–family life conflict, perhaps indicating inflexibility in the public sector. These results may be linked to those reported in [23,24,43,44] with regard to endorsing the importance of support offered to working carers in the workplace. Navigating work and care can force working carers to consider changing their work environment if the current one does not support their current family situation and care responsibilities.

On top of flexible work, access to longer care leave also helps to reduce the work–care conflict associated with withdrawal from the labour market [45]. The aforementioned European Union Directive [29] on work–life balance for parents and carers, which entered into force on 20 April 2017, also contributes to reducing the work–care conflict associated with withdrawal from the labour market. In June 2019, the Directive introduced the right to 5 days of a carer leave per year for employed workers providing personal care or support to a relative or a person living in the same household and the right to request flexible working arrangements for working carers. Although the care needs faced by working carers often require more than 5 days off work and many EU Member States have already introduced measures going beyond these provisions, the Directive can still be seen as an important step in recognising the challenges of combining work and care and in acknowledging working carers [2].

In addition to support for working carers in the workplace, the development of the LTC sector may also play an important role in supporting working carers in their working and caring roles. Our findings confirm that increasing the use of supplemental care to ease family care decreases work–family life conflict when we consider home help services but increases work–family life conflict when we consider nursing care services or care paid for in the grey market. The results are not surprising, as paying for extra care in the grey market increases the financial demands on families and such care far less regulated than care organised under the umbrella of public or private organisations. Home help services can substantially relieve the burden of care for working carers. In our view and according to our own previous research [46,47,48], when combining family care and formal care, the use of nursing care services indicates substantial care needs and heavy care demands on working carers. The last variable in the care-related block is the subjective evaluation of LTC services in the country; it is hardly surprising that as the quality of LTC services increases, work–family life conflict significantly decreases. It is also no surprise that with respect to the availability and usage of all types of LTC, from additional informal care to paid care in the grey market, those of home help and home nursing care are highest in fully fledged care regimes and the lowest in residual care regimes.

It remains uncertain how organisations can best support working carers, although some empirical studies evaluating either policies [49,50,51] or interventions [43,52,53,54] exist and show positive results. On one hand, many countries have adopted more flexible work arrangements, like remote work. On the other hand, caregiving burdens have intensified for many caregivers due to the pandemic’s strain on healthcare services, the closure of formal care services, and the consequential blurring of work–life boundaries [55]. Furthermore, the pandemic has led to an increase in the number of new caregivers [56] and to heightened mental and physical burdens on carers [57,58,59], especially in women caregivers [60]. The pandemic has highlighted the need for a better support system for caregivers and the provision of educational and emotional resources [61]. That is why the policies and workplace practices recommended in this study are even more important now, in the post-COVID landscape. Not only are the health and social care sectors important in sustaining care for older adults and promoting aging within the community; support for family carers in active employment is also essential for age-friendly societies, contributing to older adults being able to stay at home longer. Workplace organisations with flexible work arrangements, safe reductions of working hours, employment protection during care, and emergency and short- or long-term leaves are measures that can help family carers continue providing care while remaining active in the labour market.

## 5. Limitations

While our study is cross-national and, thus, expands previous research on work–family life conflict, several limitations arise from using a secondary dataset. First, while the data are not recent, given the COVID-19 period when social distancing prevented face-to-face data collection for most cross-national research, the dataset remains valid in terms of comprising the most recent sets of important variables in the study work–family life conflict. Furthermore, the data we used were collected pre-COVIDE, therefore not considering the changes in the dynamics of work and caregiving that were brought on by the pandemic. Using secondary data present other limitations, such as a sub-optimal set of questionnaire items. With respect to work–family life conflict, the index is not perfectly balanced, since it contains only one item measuring spillover from family life to the work environment, whereas there are two items assessing spillover from the work domain to the family life domain. The small number of items in the index is another shortcoming. Secondly, the EQLS study conducted in 2016 focused on the quality of LTC services, which adds a lot in terms of providing an array of items assessing the use and quality of LTC services in participating countries. Items are presented to measure the frequency of caring for disabled or infirm family members, neighbours, or friends aged under 75 and the frequency of caring for disabled or infirm family members, neighbours, or friends aged 75 or over, and we computed an index combining both items. One item measures hours of care, although it was not measured for all respondents (only beyond a certain care frequency threshold), considerably reducing the number of respondents and causing us to exclude this variable from our analysis. Certainly, there is more to care than frequency and hours, such as the amount and type of care needs, as well as the subjective burden of care. We would like the dataset to be even more inclusive of care-related items than it already is.

On the other hand, data from EQLS 2012 includes several items measuring the use of work flexibility measures, which were omitted in EQLS 2016. As a consequence, the work context is not captured at the organisational level, and we can only assume that care regimes, in some way, cover at least part of the picture of the work environment. Beyond work flexibility arrangements, the actual workplace situation, such as whether the supervisor, HRM, and co-workers are supportive, can ease or hinder the uptake of either national policy measures or organisational policy measures. These are all missing parts of the very complex research topic that were omitted in the current study.

In contrast, Ekezie et al. [55] explored the association between the mental health of older care recipients and work strain in their children who provide care and established a positive association indicating that the mental health status of the care recipient may be important when observing the burden of the care.

## 6. Conclusions

As the population ages, most care for dependent older people is provided by family members, friends, and neighbours, many of whom are also working. Care demands and work demands, as we found in our paper, contribute to work–family conflict. While caregiving, as such, is not necessarily connected to work–family life conflict, the burden of care in hours and the subjective burden of care, alongside work demands such as working hours, the obligation to replace missing hours, the fear of losing one’s job, and years spent at work, increase work strain and work–family life conflict. Gender and health also shape work-related strain, with women particularly affected by caregiving duties and financial pressures. This analysis supports the expectation that in countries where working carers are fully supported by nationally adopted policies, their workplace participation can bear higher levels of work–family life conflict. Our study brings together factors contributing to work–life conflict in a cross-national and care-regime context in a robust, hierarchical linear regression model, extending previous studies mostly conducted in single-country national contexts.

The findings of this research have significant implications for policy design and workplace practices. Governments should support caregivers by implementing comprehensive caregiving legislation, such as paid leave for those who need to provide care without facing financial difficulties, and ensure job security guarantees that can protect carers who need to take (extended) time off from losing their jobs. Another way to support caregivers is by securing flexible work arrangements, such as flexible working hours and remote work to help balance caregiving and work demands. Additionally, governments can support carers by investing in long-term care and ensuring that services like home care and institutional care are accessible, affordable, and of high quality, thereby reducing the care burden. Tax deductions for caregiving expenses can further alleviate the financial strain on families. Since the majority of caregivers are women, governments should introduce gender-sensitive policies like additional leave benefits. These accommodations can reduce the stress and job strain among caregivers, improve their well-being, and enhance their participation in the labour market.

For the next European Quality of Life Survey, we suggest using a more balanced measurement tool to assess work–life conflict. This tool should place equal emphasis on both work-to-family and family-to-work spillover, as the current tool uses only one item to measure the impact of family life on work, while two items focus on the influence of work on family life. This imbalance may limit the ability to accurately assess both directions of the conflict. Another issue is the lack of organizational-level variables in this study, such as support from supervisors or colleagues and human resources policies. Furthermore, to better understand the work–life conflict, we recommend including data on flexible work arrangements, the types and complexity of caregiving tasks, and the subjective psychological burden experienced by caregivers. Finally, incorporating qualitative insights or case studies could further enhance our understanding of the personal experiences behind the data on work–life conflict.

## Figures and Tables

**Table 1 healthcare-12-02415-t001:** Independent variables of control and country context.

Variable	Value
Gender	0, Female; 1, male
Age	Years
Marital status	0, Not married; 1, married
Household size	Number
Health	In general, how is your health? (1, very good; 2, good; 3, fair; 4, bad; 5, very bad)
Urbanisation (subjective)	1, Open countryside; 2, a village/small town; 3, a medium to large town; 4, a city or city suburb
Partial care regime (PCR)	PCR with short-term leave entitlements and protection for working carers
Residual care regime (RCR)	RCR mainly based on flexible work (time) organisation only

**Table 2 healthcare-12-02415-t002:** Care-related items.

Variable	Value
Frequency of care	(Frequency of caring for disabled or infirm family members, neighbours, or friends aged under 75 + frequency of caring for disabled or infirm family members, neighbours, or friends aged 75 or over)/2 (1, never; 2, less often; 3, once or twice a week; 4, several days a week; 5, every day)
Informal care	Care by family members, friends, or neighbours free of charge (0, no; 1, yes)
Paid care	Paid care by someone outside of formal health and care services (0, no; 1, yes)
Nursing care services	Used nursing care services at your/this person’s home (0, no; 1, yes)
Home help services	Used home help or personal care services in your/this person’s home (0, no; 1, yes)
Quality of LTC services	Quality of long-term care services (1, very poor quality … 10, very high quality)

**Table 3 healthcare-12-02415-t003:** Work-related items.

Variable	Value
Sector	0, Not public; 1, public
Losing one’s job	Likelihood of losing one’s job in the next 6 months (1, very unlikely; 2 rather unlikely; 3, neither unlikely nor likely; 4, rather likely; 5, very likely)
Hours spent at main job	In hours
Minutes commuting	Minutes spent commuting
Sufficiency of income in old age	How worried are you that your income in old age will not be sufficient? (1, not worried at all … 10, extremely worried)
Making ends meet	Making ends meet (1, very easily; 2, easily; 3, fairly easily; 4, with some difficulty; 5, with difficulty; 6, with great difficulty)

**Table 4 healthcare-12-02415-t004:** Results of the hierarchical regression analysis.

Variable	1			2			3		
	Unst. C.		Std. C.	Unst. C.		Std. C.	Unst. C.		Std. C.
	B	Std. Error	Beta	B	Std. Error	Beta	B	Std. Error	Beta
Constant	2.519	0.072		2.808	0.083		0.872	0.102	
Gender	−0.069	0.023	**−0.030**	−0.064	0.023	**−0.028**	−0.172	0.023	**−0.076**
Age	−0.011	0.001	**−0.112**	−0.012	0.001	**−0.115**	−0.009	0.001	**−0.085**
Marital status	0.007	0.026	0.003	0.010	0.026	0.004	0.048	0.025	**0.021**
Household size	0.073	0.010	**0.082**	0.072	0.010	**0.081**	0.057	0.009	**0.064**
Health	0.331	0.016	**0.216**	0.317	0.016	**0.207**	0.207	0.016	**0.136**
Urbanisation (subjective)	−0.002	0.012	−0.002	0.002	0.012	0.001	−0.001	0.012	−0.001
Partial care regime	0.100	0.036	**0.030**	0.071	0.036	**0.021**	−0.137	0.035	**−0.041**
Residual care regime	0.174	0.026	**0.072**	0.157	0.026	**0.065**	−0.061	0.026	**−0.025**
Frequency of care				0.016	0.007	**0.025**	0.021	0.006	**0.033**
Informal care				0.029	0.045	0.008	0.060	0.043	0.016
Paid care				0.159	0.063	**0.029**	0.109	0.060	**0.020**
Nursing care services				0.184	0.059	**0.043**	0.103	0.055	**0.024**
Home help services				−0.283	0.063	**−0.060**	−0.210	0.060	**−0.044**
Quality of LTC services				−0.050	0.006	**−0.092**	−0.027	0.005	**−0.050**
Sector							0.047	0.024	**0.019**
Losing one’s job							0.113	0.011	**0.106**
Hours spent at main job							0.026	0.001	**0.194**
Minutes commuting							0.002	0.000	**0.061**
Sufficiency of income							0.020	0.005	**0.046**
Making ends meet							0.195	0.010	**0.206**
Sig. F change	*p* < 0.001			*p* < 0.001			*p* < 0.001		
R^2^	0.064			0.076			0.177		
Adjusted R^2^	0.063			0.075			0.175		

**Table 5 healthcare-12-02415-t005:** Averages of all work-related independent variables across country regimes.

Variable	Fully Fledged CR	Partial CR	Residual CR	Total
W-C conflict	2.8202	2.9790	3.0636	2.9227
Gender	0.53	0.56	0.53	0.53
Age	42.0444	43.2140	40.5065	41.6395
Marital status	0.53	0.64	0.64	0.58
Household size	2.8198	3.0093	3.2032	2.9741
Health	1.90	2.01	1.94	1.93
Urbanisation (subjective)	2.65	2.60	2.76	2.68
Frequency of care	3.0389	2.8384	2.6427	2.8783
Informal care	0.1255	0.0812	0.0683	0.1006
Paid care	0.0527	0.0431	0.0209	0.0405
Nursing care services	0.0866	0.0477	0.0357	0.0645
Home help services	0.0712	0.0475	0.0301	0.0542
Quality of LTC services	6.4477	5.6066	5.8040	6.1154
Sector	0.32	0.27	0.27	0.30
Fear of losing one’s job	1.73	2.11	2.09	1.89
Hours spent at main job	38.0458	39.5600	41.7609	39.5055
Minutes spent commuting	43.0848	38.0726	37.5206	40.6021
Sufficiency of income	5.63	6.58	6.59	6.07
Making ends meet	2.73	3.48	3.24	2.99

## Data Availability

European Foundation for the Improvement of Living and Working Conditions (Eurofound). European Quality of Life Survey Integrated Data File, 2003–2016. 2023. Available online: http://doi.org/10.5255/UKDA-SN-7348-3 (accessed on 11 June 2024).

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
