# Peer review of "Working Carers in Europe and How Their Caring Responsibilities Impact Work–Family Life Conflict: Analysis of the European Quality of Life Survey"

_healthcare, 2024, doi:10.3390/healthcare12232415_

Round 1

Reviewer 1 Report

Comments and Suggestions for Authors

There are some revisions that I would like the authors to address and/or to consider

1.    Please provide evidence for normality of the data

2.    Please Present and tabulate sample characteristics

3.    Please report reliability for measurements

4.    p=.000, not meaning. Please report statistical forms such as: p<.001

5.    Please theoretically justify why hierarchical regression analysis has been used

6.    Please provide the practical implications of the study

Author Response

10/10/2024

Reviewer 1-Revisions

Article: Working carers in Europe and how their caring responsibilities impact work–family life conflict: Analysis of the European Quality of Life Survey

  1. Please provide evidence for normality of the data.

Thank you for the request. We have not included the graphs in the manuscript as we did not observe this as a standard in the journal. Here is the plot of the dependent variable and the P-P plot of residuals.

  1. Please Present and tabulate sample characteristics.

Thank you for the suggestion! The following text has been included as the first paragraph of the results (R 233 onwards).

The average level of work-life conflict was 3.02. The analysed sample consisted of 45.4% female respondents, of average age 41 years, 58.3% being married, living in households of 3 people on average. In general, health was evaluated at 1.94. 7.6% live in open countryside, 38.1% in a village or small town; 23.1% in a medium to large town, and 31.2% in a city or city suburb. 13.0%% lives in PCR with short-term leave entitlements and protection for working carers, whereas 34.0% lives in RCR mainly based on flexible work (time) organisation only.

Frequency of care was evaluated at 3.00 on average, 9.6% received care by family members, friends, or neighbours free of charge, 3.9% received paid care by someone outside the formal health and care services, 6.0% used nursing care services at your/this person's home, 5.2% used home help or personal care services in your/this person's home. The quality of long-term care services was estimated at 6.15 on average.

29% are working in the public sector, with an average 1.98 likelihood of losing one’s job in the next 6 months, the majority (81%) are working on permanent contracts with 40.44 working hours on average. Average minutes of commuting were 40.32. On average, fear that income in old age will not be sufficient, was evaluated at 1.91, and the ease of making ends meet was estimated at 3.00 on average. 

  1. Please report reliability for measurements.

Thank you for the suggestion! The following sentence was included in section 2. Material and Metods (R 214): Cronbach’s Alpha was .784.

  1. p=.000, not meaning. Please report statistical forms such as: p<.001

Thank you very much for the suggestion! Duly corrected in the text below table 1.

  1. Please theoretically justify why hierarchical regression analysis has been used.

Thank you for the suggestion. The following sentence has been inserted in the methods section (R225 onwards).

Given that care characteristics and work situation conceptually contribute to work-life conflict as two separate sources, apart from control variables described in the introduction, the hierarchical regression method was used.

  1. Please provide the practical implications of the study.

Thank you very much for this suggestion! The practical implications of the study have been added under the Conclusion section (R477–R491).

The findings of this research have significant implications for policy design and workplace practices. Governments should support caregivers by implementing comprehensive caregiving legislation, such as paid leave for those who need to provide care, without facing financial difficulties, and ensure job security guarantees that would protect carers who need to take (extended) time off from losing their jobs. Another way to support caregivers is by securing flexible work arrangements, such as flexible working hours and remote work to help balance caregiving and work demands. Additionally, governments can support carers by investing in long-term care and ensuring that services like home care and institutional care are accessible, affordable, and high-quality, thereby reducing the care burden. Tax deductions for caregiving expenses can further alleviate the financial strain on families. Since the majority of caregivers are women, governments should introduce gender-sensitive policies, like additional leave benefits. These accommodations can reduce the stress and job strain among caregivers, improve their well-being, and enhance their participation in the labour market.

Reviewer 2 Report

Comments and Suggestions for Authors

First of all, I would like to thank the authors for the opportunity to read this paper. The subject is both interesting and engaging, however, there are a few aspects that the authors should address, in order to enhance the quality of the paper.

Thus, the Introduction section provides a too sudden presentation of the statistical data, without making an easy passage to these aspects. At least an introductory phrase would be welcome, in order to familiarize the reader with the subject, given that not everyone is familiar with it.

The article analyzes how informal caregiving for dependent individuals influences the conflict between work and family life. It focuses on the interaction between caregiving responsibilities and work demands and the introductory section should offer an overall image of the main concepts that will be discussed in the content. This is necessary also because the article does not provide a distinct Literature review section, where the main relevant and recent sources from literature to be analysed.

The study highlights the idea that the aging population puts pressure on care systems, with most caregiving responsibilities falling on family members, friends, and neighbors.

The strengths of this article are its rigorous analysis and the use of relevant data, but some limitations are also highlighted, such as the limited set of variables related to work-life conflict and the lack of organizational measures regarding flexible work.

The article, however, presents several weaknesses:

We can mention here the limitations of the dataset. The present study uses secondary data from 2016, meaning the data is not very recent. The authors point out that although these are the most recent data available at the time of research, they do not reflect the significant changes in recent years, particularly those caused by the COVID-19 pandemic​. Using more recent data is recommended, as that current one is almost a decade old.

We can also indicate the incomplete measurement tools, as the index used to measure work-life conflict is unbalanced. It has only one item measuring the influence of family life on work, while two items focus on the influence of work on family life. This could limit the ability to accurately assess both directions of the conflict.

The lack of organizational variables can also be an issue. Thus, although the study analyzes the impact of caregiving on work-life conflict, it does not include important organizational-level variables, such as support from supervisors or colleagues and human resources policies. These elements would have been useful for better assessing how the work environment influences this conflict.

Also, the study does not include data on flexible work arrangements, although these could have a significant impact on work-life conflict for caregivers. The article examines the frequency and hours of caregiving, it does not, however, delve deeply into other important aspects, such as the types and complexity of caregiving tasks or the subjective psychological burden experienced by caregivers​.

These limitations somewhat reduce the practical value and applicability of the article's conclusions, especially in terms of fully understanding the dynamics of work-life conflict for caregivers.

A distinct Conclusions section should be included, while the References should be also updated, as 29 sources of 55 are older that 2020.

Author Response

10/10/2024

Reviewer 2-Revisions

Article: Working carers in Europe and how their caring responsibilities impact work–family life conflict: Analysis of the European Quality of Life Survey

  1. The Introduction section provides a too sudden presentation of the statistical data, without making an easy passage to these aspects. At least an introductory phrase would be welcome, in order to familiarize the reader with the subject.

Thank you for the suggestion! An introductory phrase section is now included in the manuscript (R34–R38).

  1. The introductory section should offer an overall image of the main concepts that will be discussed in the content. This is necessary also because the article does not provide a distinct Literature review section, where the main relevant and recent sources from literature to be analysed.

Thank you for the suggestion! An overall image of the main concepts that will be discussed in the content is now included in the manuscript (R38–R44).

  1. The present study uses secondary data from 2016, meaning the data is not very recent. The authors point out that although these are the most recent data available at the time of research, they do not reflect the significant changes in recent years, particularly those caused by the COVID-19 pandemic. Using more recent data is recommended, as that current one is almost a decade old.

Thank you for your observation. We are unable to use a later data set as in fact, the EQLS survey was conducted at the latest in 2016. We are including a print screen of the series to show that our data set is the latest (https://www.eurofound.europa.eu/en/surveys/european-quality-life-surveys-eqls). 

Some data collection was indeed collected after 2016, however two surveys were combined and shortened. Therefore, the selection of variables presented at later dates was not sufficient for analysis as performed in the manuscript. Description of the survey is as follows (https://www.eurofound.europa.eu/en/surveys/living-and-working-eu-e-survey/e-survey-methodology):

The e-survey includes questions related to the pandemic’s impact on people’s lives and goes further to investigate living and working conditions in a post-pandemic world, taking into account the uncertain reality caused by the war in Ukraine, record-high inflation, and sharp rises in the cost of living. The e-survey asks questions about respondents’ employment situation, work-life balance, and their use of telework both during and after the COVID-19 crisis.  

It also examines the quality of life and society, with questions ranging from life satisfaction, happiness, and optimism, to health and levels of trust in institutions, job quality, and health and safety at work. The e-survey also looks at the life of young people in terms of traineeships and apprenticeships, skills and access to training, and their hopes and plans for the future. 

To make things worse, the probability sampling was not undertaken, thus making inferential statistics out of reach for high quality statistical analysis (https://www.eurofound.europa.eu/en/surveys/living-and-working-eu-e-survey/e-survey-methodology):

Unlike Eurofound’s two regular population surveys – the European Working Conditions Survey (EWCS) and the European Quality of Life Survey (EQLS) – the e-survey applied a non-probability sampling method. Survey participants were recruited using primarily social media advertisements, complemented with snowball sampling. This method produces a non-representative sample.

Although large segments of the population have access to the internet, those with no access were by default excluded from the sample. Internet penetration levels vary by country and are lower among certain segments of the population, notably elderly people, those living in remote areas and people with low education. Taking part in an online survey also requires digital literacy. Additionally, people who do not use any of the social media channels, where the e-survey was promoted, will not have exposure to the survey, unless they receive it from other participants via snowballing. It is not possible to correct for the bias that is introduced by these factors.

Respondents who agreed to participate in later rounds of the e-survey were asked to submit their email address (this data was pseudonymised and cannot be linked to their survey data), and were invited to participate in subsequent survey rounds. Over 50,000 panel members participated in multiple rounds of the e-survey.

  1. We can also indicate the incomplete measurement tools, as the index used to measure work-life conflict is unbalanced. It has only one item measuring the influence of family life on work, while two items focus on the influence of work on family life. This could limit the ability to accurately assess both directions of the conflict.

Thank you for this observation. As we have duly reported in the limitations section, we have no control over what is included in the EQLS questionnaire, given that this is a secondary data set. We would be happy if the next round of EQLS included more items. For now, we included it as a suggestion for future research under the Conclusion section (R492 –R497).

  1. The lack of organizational variables can also be an issue. Thus, although the study analyzes the impact of caregiving on work-life conflict, it does not include important organizational-level variables, such as support from supervisors or colleagues and human resources policies.

Thank you for this observation. As we have duly reported in the limitations section, we have no control over what is included in the EQLS questionnaire, given that this is a secondary data set. We would be happy if the next round of EQLS included more items, especially organizational-level variables, such as support from supervisors or colleagues and human resources policies. For now, we included it as a suggestion for future research under the Conclusion section (R497–R498).

  1. The study does not include data on flexible work arrangements, although these could have a significant impact on work-life conflict for caregivers. The article examines the frequency and hours of caregiving, it does not, however, delve deeply into other important aspects, such as the types and complexity of caregiving tasks or the subjective psychological burden experienced by caregivers.

Thank you for this observation. As we have duly reported in the limitations section, we have no control over what is included in the EQLS questionnaire, given that this is a secondary data set. We would be happy if the next round of EQLS included more items, such as the types and complexity of caregiving tasks or the subjective psychological burden experienced by caregivers. For now, we included it as a suggestion for future research under the Conclusion section (R498–R501)

  1. A distinct Conclusions section should be included.

Thank you for the suggestion! Conclusions section is now included in the manuscript (R467 onwards)

  1. The References should be also updated, as 29 sources of 55 are older that 2020.

Thank you for this observation! Given that we have now expanded the manuscript with the Conclusions section, new references (related to the Covid period and afterward), and 6 new references related to this period and post Covid period were added. Thus, the balance between pre 2020 and post 2020 publications, is now improved.

Reviewer 3 Report

Comments and Suggestions for Authors

The findings are highly significant for policymakers and organizations, emphasizing the need for supportive workplace policies to reduce work–family conflicts. The study highlights the critical role of care regimes and demonstrates that national and organizational support systems can help mitigate stress and conflict for working carers. These insights could inform the development of future care policies, particularly in countries with underdeveloped care systems. Additionally, the identification of key predictors such as gender, health, care demands, and job-related factors enhances the impact of the research, offering valuable direction for targeted interventions.

While the statistical analysis is robust, addressing a few areas could further strengthen the manuscript, such as ensuring balanced measurement of work-to-family and family-to-work spillover, incorporating more recent data, and expanding on the policy recommendations for organizations and governments:

1. Balance in Measurement: The work–life conflict index could benefit from a more equal emphasis on both work-to-family and family-to-work spillover.

2. Dataset Limitations: Address the potential impact of using pre-pandemic data and consider referencing more recent studies that highlight changes in work and care dynamics, especially post-COVID-19.

3. Qualitative Data: While the quantitative analysis is robust, incorporating qualitative insights or case studies could enhance understanding of the personal experiences behind the data.

4. Policy Implications: The implications section could be expanded to provide more specific recommendations for organizations and governments to support working carers.

Overall, the paper is a strong contribution to the literature, and addressing these points would make it even stronger.

Author Response

10/10/2024

Reviewer 3-Revisions

Article: Working carers in Europe and how their caring responsibilities impact work–family life conflict: Analysis of the European Quality of Life Survey

  1. While the statistical analysis is robust, addressing a few areas could further strengthen the manuscript, such as ensuring balanced measurement of work-to-family and family-to-work spillover (limitations), incorporating more recent data (limitations), and expanding on the policy recommendations for organizations and governments.

Thank you for your observation. We are unable to use a newer  data set as the EQLS survey was conducted at the latest in 2016. We are including a print screen of the series to show that our data set is the latest (https://www.eurofound.europa.eu/en/surveys/european-quality-life-surveys-eqls). 

Some data collection was indeed conducted after 2016, however two surveys were combined and shortened. Therefore, the selection of variables presented at later dates was not sufficient for analysis as performed in the manuscript. Description of the survey is as follows (https://www.eurofound.europa.eu/en/surveys/living-and-working-eu-e-survey/e-survey-methodology):

The e-survey includes questions related to the pandemic’s impact on people’s lives and investigates living and working conditions in a post-pandemic world, dealing with the uncertain reality caused by the war in Ukraine, record-high inflation, and sharp rises in the cost of living. The e-survey asks questions about respondents’ employment situation, their work-life balance, and their use of telework both during and after the COVID-19 crisis. 

It also examines the quality of life and society, with questions ranging from life satisfaction, happiness, and optimism, to health and levels of trust in institutions, job quality, and health and safety at work. The e-survey also looks at the life of young people in terms of traineeships and apprenticeships, skills and access to training, and their hopes and plans for the future.

To make things worse, the probability sampling was not undertaken, thus making inferential statistics out of reach for high quality statistical analysis (https://www.eurofound.europa.eu/en/surveys/living-and-working-eu-e-survey/e-survey-methodology):

Unlike Eurofound’s two regular population surveys – the European Working Conditions Survey (EWCS) and the European Quality of Life Survey (EQLS) – the e-survey applied a non-probability sampling method. Survey participants were recruited using primarily social media advertisements, complemented by snowball sampling. This method produces a non-representative sample.

Although large segments of the population have access to the internet, those with no access were by default excluded from the sample. Internet penetration levels vary by country and are lower among certain segments of the population, notably elderly people, those living in remote areas, and people with low education. Taking part in an online survey also requires digital literacy. Additionally, people who do not use any social media channels, where the e-survey was promoted, will not have exposure to the survey, unless they receive it from other participants via snowballing. It is not possible to correct for the bias that is introduced by these factors.

Respondents who agreed to participate in later rounds of the e-survey were asked to submit their email addressess (this data was pseudonymised and cannot be linked to their survey data) and were invited to participate in subsequent survey rounds. Over 50,000 panel members participated in multiple rounds of the e-survey.

We addressed the policy recommendations for organizations and governments in the manuscript (R478–R491).

  1. Balance in Measurement: The work–life conflict index could benefit from a more equal emphasis on both work-to-family and family-to-work spillover.

Thank you for this observation. As we have duly reported in the limitations section, we have no control over what is included in the EQLS questionnaire, given that this is a secondary data set. We would be happy if the next round of EQLS included more items. For now, we included it as a suggestion for future research under the Conclusion section (R492–R497).

  1. Dataset Limitations: Address the potential impact of using pre-pandemic data and consider referencing more recent studies that highlight changes in work and care dynamics, especially post-COVID-19.

We addressed the potential impact of using pre-pandemic data and referencing more recent studies that highlight changes in work and care dynamics, especially post-COVID-19 in the manuscript under the Conclusion section (R503–R514).

  1. Qualitative Data: While the quantitative analysis is robust, incorporating qualitative insights or case studies could enhance understanding of the personal experiences behind the data.

Thank you very much for your suggestion! We will keep that in mind for future research. For now, we included it as a suggestion for future research under the Conclusion section (R501–R503).

  1. Policy Implications: The implications section could be expanded to provide more specific recommendations for organizations and governments to support working carers.

Thank you very much for your suggestion! We included your suggestion under the Conclusion section (R478–R491).
